## [Decision Letter · Decision Letter 0]

19 Mar 2025

Dear Dr. Shimizu,

Thank you for submitting your manuscript to PLOS ONE. After careful consideration, we feel that it has merit but does not fully meet PLOS ONE’s publication criteria as it currently stands. Therefore, we invite you to submit a revised version of the manuscript that addresses all the points raised during the review process.

We look forward to receiving your revised manuscript.

Kind regards,

Mária A. Deli, M.D., Ph.D.

Academic Editor

PLOS ONE

Journal Requirements:

“This work was supported by the Joint Research Program of the Bio-signal Research Center, Kobe University (291004 to TS), and a Grant-in-Aid for Scientific Research (C) from the Japan Society for the Promotion of Science (Grant Number 21K09142 to TS).”

“This work was supported by the Joint Research Program of the Biosignal Research Center, Kobe University (291004 to TS), and a Grant-in-Aid for Scientific Research (C) from the Japan Society for the Promotion of Science (Grant Number 21K09142 to TS). We thank N. Tsuchiya, a student in our laboratory, for assistance with the cell assay. We would like to thank Editage (www.editage.jp ) for English language editing.”

“This work was supported by the Joint Research Program of the Bio-signal Research Center, Kobe University (291004 to TS), and a Grant-in-Aid for Scientific Research (C) from the Japan Society for the Promotion of Science (Grant Number 21K09142 to TS).”

Reviewers' comments:

Reviewer's Responses to Questions

**Comments to the Author**

1. Is the manuscript technically sound, and do the data support the conclusions?

Reviewer #1: Yes

Reviewer #2: Partly

Reviewer #3: Yes

2. Has the statistical analysis been performed appropriately and rigorously?

Reviewer #1: Yes

Reviewer #2: Yes

Reviewer #3: Yes

3. Have the authors made all data underlying the findings in their manuscript fully available?

Reviewer #1: No

Reviewer #2: Yes

Reviewer #3: Yes

4. Is the manuscript presented in an intelligible fashion and written in standard English?

Reviewer #1: Yes

Reviewer #2: No

Reviewer #3: Yes

Reviewer #1: All relevant data are not within the manuscript and its Supporting Information files. Nor on other options.

The authors explored ARP and its metabolite OPC-14857 (OPC), inhibitory effects on cell proliferation using glioblastoma cell 23 lines (U-251, T98G, and U-87 cells as well as their effects on the cell cycle, cytoskeleton, and cell migration.

The manuscript is well written, clear with adequate conclusions. Results showed that OPC have anticancer effects and suggesting use of ARP for drug repurposing in glioblastoma.

Some minor changes are necessary.

Abstract

Distinct effects have to be detailed, it is too general. Line 29

Discussion

To my view, discussion about doses used in vitro in the manuscript and those used in vivo during glioblastoma therapy has to be discussed, especially some calculation that consider pharmacokinetics of ARP. It will more elaborate possible use of ARP in glioblastoma therapy. Moreover, relevance to current treatment and drug is also needed.

Figure 2. D) ANOVA and t-test has to be presented.

Fig 4. Regrouping will be better. Left side of graph comparison of DMSO together, right side ARP for A), the same for OPC B).

Reviewer #2: This manuscript suggests that aripiprazole main metabolite OPC-14857 has anticancer activity in malignant glioblastoma. There are a few points that should be addressed and corrected throughout the text.

1. The authors needs to separate the methods, results, and figure legends. For example, the Figure 1 legend is included in the introduction part.

2. The authors need to describe the experiment methods more in detail.

3. The subtitle is Water-soluble tetrazolium assay, but use the more general words for the cell viability assay. Cell viability assay, MTT assay, WST assay, CCK-8 assay, etc. And please explain experimental condition in detail, for example, the incubation time of Cell Counting Kit-8 in cell viability assay.

4. This OPC-14857 is originally an active metabolite of aripiprazole, antipsychotic drug, of which therapeutic window is so narrow and the cellular concentration is so low. The concentration used in this study is 30 uM which may be toxic to normal cell and could show the side effect. Is that concentration available in human plasma in clinics? Did you check the cell viability in normal cells with this concentration? Or how about safety in animal? They need to explain the high centration issue.

5. In single treatments, they used 30uM , but cominaiton with DOX, they used 5uM. It would be fine if they used the same concentration.

6. Is there any results using animal model (xenograft) or organoid?

7. In Introduction part, they mentioned “Treatment with ARP-induced anticancer effects in glioblastoma, breast cancer, gastric adenosquamous carcinoma, and colon carcinoma cells has been reported [9–11].” It needs to add more recent paper, e.g., Cho et al., Repositioning of aripiprazole, an antipsychotic drug, to sensitize the chemotherapy of pancreatic cancer, 2025.

Reviewer #3: This study investigates the anticancer effects of aripiprazole (ARP) and its primary metabolite OPC-14857 (OPC) against malignant glioblastoma. The study is well-structured and provides comprehensive in vitro data supporting the comparable anticancer potential of OPC and ARP, along with synergistic effects with doxorubicin. The focus on drug metabolites is valuable and underexplored in cancer research. However, several areas require improvement to enhance the manuscript's scientific rigor and clarity. Addressing the above points will significantly enhance its clarity, impact, and readiness for publication.

1. The study addresses the important issue of glioblastoma treatment and the need for alternative or adjunct therapies. In addition, the experimental design is logical, utilizing multiple glioblastoma cell lines and assessing proliferation, migration, cytoskeleton alterations, and cell cycle effects. However, the combinational effects with doxorubicin are particularly relevant for translational implications. Authors added the issues in the Introduction part.

2. The study is limited to in vitro findings. While this is acknowledged, adding even preliminary in vivo data or discussing potential models and limitations would strengthen the conclusions.

3. The paper discusses differences in F/G-actin ratio and cell cycle arrest but does not delve into potential signaling pathways (e.g., Src, PI3K/Akt, or p53 involvement). Authors might be performed the Western blot or RT-PCR analysis to provide depth to the mechanistic interpretation.

4. Statistical analysis performed again. The sample sizes (n=3 or n=4) are standard but rather small for robust conclusions. Increasing replicates or clarifying if experiments were performed in triplicate with separate biological repeats would be advisable.

5. While the manuscript states that ARP and OPC are more effective than temozolomide (TMZ), a direct statistical comparison and dose equivalency discussion are missing. Therefore, authors added clarifying clinical relevance would enhance impact.

6. The combined effect of DOX and ARP/OPC is shown, but synergy quantification (e.g., Combination Index or Bliss analysis) would provide stronger evidence.

7. The manuscript would benefit from additional language polishing for fluency and scientific tone.

8. Ensure all figures are high-resolution with clear legends. The figure captions should fully explain the experimental conditions.

9. Please define all abbreviations at first mention, including IM-54.

10. Explain the rationale for choosing specific ARP and OPC concentrations, especially in combination studies.

11. Some references are repeated multiple times with different links. Ensure consistency and proper formatting.

**Do you want your identity to be public for this peer review?** For information about this choice, including consent withdrawal, please see our Privacy Policy

Reviewer #1: No

Reviewer #2: No

Reviewer #3: No

---

## [Author Response · Author response to Decision Letter 1]

10 Jul 2025

We appreciate the constructive feedback from the editor and reviewers. We have revised the manuscript accordingly and included new data as requested. A detailed response to each comment is provided in the attached Response to Reviewers document.

For your reference, we have also included the same response to reviewers below.

Reviewer #1: All relevant data are not within the manuscript and its Supporting Information files. Nor on other options.

(Reply)

Thank you for your comment. We respectfully believe that all relevant data have been included in the manuscript and its Supporting Information files. If there are any specific data or sections that appear to be missing or unclear, we would be grateful for further clarification and will address them accordingly.

The authors explored ARP and its metabolite OPC-14857 (OPC), inhibitory effects on cell proliferation using glioblastoma cell lines (U-251, T98G, and U-87 cells as well as their effects on the cell cycle, cytoskeleton, and cell migration. The manuscript is well written, clear with adequate conclusions. Results showed that OPC have anticancer effects and suggesting use of ARP for drug repurposing in glioblastoma.

Some minor changes are necessary.

Abstract

Distinct effects have to be detailed, it is too general. Line 29

(Reply)

Thank you for your valuable comment. We have revised the Abstract to specifically highlight the differences in the effects of ARP and OPC. (line 27-34)

Discussion

To my view, discussion about doses used in vitro in the manuscript and those used in vivo during glioblastoma therapy has to be discussed, especially some calculation that consider pharmacokinetics of ARP. It will more elaborate possible use of ARP in glioblastoma therapy. Moreover, relevance to current treatment and drug is also needed.

(Reply)

Thank you for your insightful comment. Regarding the maximum plasma concentrations (Cmax), ARP and OPC reach 69 nM and 6.5 nM, respectively. In contrast, the in vitro IC50 values for both ARP and OPC were approximately 20 µM in U251, T98G, and U87 cells. This indicates a considerable discrepancy between the in vitro effective concentrations and the achievable plasma concentrations, suggesting that immediate clinical application may be challenging. However, we believe that the findings of this study could serve as a foundation for further development of these compounds as seed compounds, with the aim of narrowing the gap between effective and clinically achievable concentrations. Moreover, ARP has a high oral bioavailability of 87% and is known to effectively cross the blood-brain barrier. Its metabolite, OPC, retains biological activity, suggesting favorable pharmacokinetic properties for both compounds as lead candidates. Notably, the in vitro activity profiles of ARP and OPC differ substantially from that of temozolomide, a DNA-alkylating agent currently used for glioblastoma treatment. Given the distinct chemical structures of ARP and OPC, it is unlikely that they act as alkylating agents, raising the possibility of a novel mechanism of action. We have added this discussion to the revised manuscript. (line 361-375)

Figure 2. D) ANOVA and t-test has to be presented.

(Reply)

Thank you for your valuable comment. As additional data have been included, the original Figure 2D has been renumbered as Figure 2E in the revised manuscript. We performed statistical analysis for Figure 2E and have clearly described the method used in the figure legend of Figure 2.

Fig 4. Regrouping will be better. Left side of graph comparison of DMSO together, right side ARP for A), the same for OPC B).

(Reply)

Thank you for your helpful comment. As suggested, we have arranged the DMSO control condition on the left and the ARP and OPC treatment conditions on the right. (Fig. 4)

Reviewer #2: This manuscript suggests that aripiprazole main metabolite OPC-14857 has anticancer activity in malignant glioblastoma. There are a few points that should be addressed and corrected throughout the text.

1. The authors needs to separate the methods, results, and figure legends. For example, the Figure 1 legend is included in the introduction part.

(Reply)

Thank you for your valuable comment. We believe that the Methods and Results sections are clearly separated; however, the figure legend for Figure 1 may have been confusing. According to the submission guidelines, “Figure captions must be inserted in the text of the manuscript, immediately following the paragraph in which the figure is first cited,” and we followed this instruction by placing the caption accordingly.

2. The authors need to describe the experiment methods more in detail.

(Reply)

Thank you for your valuable comment. We have added further experimental details to the Materials and Methods section.

3. The subtitle is Water-soluble tetrazolium assay, but use the more general words for the cell viability assay. Cell viability assay, MTT assay, WST assay, CCK-8 assay, etc. And please explain experimental condition in detail, for example, the incubation time of Cell Counting Kit-8 in cell viability assay.

(Reply)

Thank you for your helpful comment. We have revised the subsection title in the Methods section and added details such as the incubation time.

4. This OPC-14857 is originally an active metabolite of aripiprazole, antipsychotic drug, of which therapeutic window is so narrow and the cellular concentration is so low. The concentration used in this study is 30 uM which may be toxic to normal cell and could show the side effect. Is that concentration available in human plasma in clinics? Did you check the cell viability in normal cells with this concentration? Or how about safety in animal? They need to explain the high centration issue.

(Reply)

Thank you for your helpful comment.

As you correctly pointed out, the maximum plasma concentrations (Cmax) in clinical settings are 69 nM for ARP and 6.5 nM for OPC. Therefore, the concentrations of ARP and OPC used in this study are considerably higher than clinically achievable levels, and immediate clinical application may be difficult. However, we believe that the findings of this study can serve as a basis for future drug discovery efforts, and the compounds may be further developed as lead compounds.

In addition, we evaluated the activity of ARP and OPC using HEK293 cells as a model for normal human cells. We have added the relevant methods, results, and discussion to the manuscript. (line84-90, 113-115, 218-222, 376-387)

We fully acknowledge the importance of animal studies. However, due to technical and resource limitations, we are currently unable to conduct additional in vivo experiments. We apologize for this. We also recognize the limitations inherent in animal models and believe that in vivo studies should be conducted with careful consideration. We have added a discussion of this issue to the revised Discussion section. (line 441-449)

5. In single treatments, they used 30uM, but combination with DOX, they used 5uM. It would be fine if they used the same concentration.

(Reply)

Thank you for your valuable comment regarding the consistency of the concentrations used in single and combination treatments. As you suggested, we initially conducted combination experiments using ARP or OPC at 30 µM, the same concentration as in the single-agent treatment. However, under these conditions, combined effects were not so remarkable. In contrast, a more evident synergistic effect was observed at 5 µM of compound A in combination with doxorubicin. Therefore, we chose to present the 5 µM combination data in the manuscript. We appreciate your understanding.

6. Is there any results using animal model (xenograft) or organoid?

(Reply)

Thank you for your helpful comment. We fully acknowledge the importance of animal studies. However, due to technical and resource limitations, we are currently unable to conduct additional in vivo experiments. We apologize for this. We also recognize the limitations inherent in animal models and believe that in vivo studies should be conducted with careful consideration. We have added a discussion of this issue to the revised Discussion section. (line 441-449)

7. In Introduction part, they mentioned “Treatment with ARP-induced anticancer effects in glioblastoma, breast cancer, gastric adenosquamous carcinoma, and colon carcinoma cells has been reported [9–11].” It needs to add more recent paper, e.g., Cho et al., Repositioning of aripiprazole, an antipsychotic drug, to sensitize the chemotherapy of pancreatic cancer, 2025.

(Reply)

Thank you for your suggestion. We have added the recommended article as a reference in the Introduction section.

Reviewer #3: This study investigates the anticancer effects of aripiprazole (ARP) and its primary metabolite OPC-14857 (OPC) against malignant glioblastoma. The study is well-structured and provides comprehensive in vitro data supporting the comparable anticancer potential of OPC and ARP, along with synergistic effects with doxorubicin. The focus on drug metabolites is valuable and underexplored in cancer research. However, several areas require improvement to enhance the manuscript's scientific rigor and clarity. Addressing the above points will significantly enhance its clarity, impact, and readiness for publication.

1. The study addresses the important issue of glioblastoma treatment and the need for alternative or adjunct therapies. In addition, the experimental design is logical, utilizing multiple glioblastoma cell lines and assessing proliferation, migration, cytoskeleton alterations, and cell cycle effects. However, the combinational effects with doxorubicin are particularly relevant for translational implications. Authors added the issues in the Introduction part.

2. The study is limited to in vitro findings. While this is acknowledged, adding even preliminary in vivo data or discussing potential models and limitations would strengthen the conclusions.

(Reply)

Thank you for your helpful comment. We fully acknowledge the importance of animal studies. However, due to technical and resource limitations, we are currently unable to conduct additional in vivo experiments. We apologize for this. We also recognize the limitations inherent in animal models and believe that in vivo studies should be conducted with careful consideration. We have added a discussion of this issue to the revised Discussion section. (line 441-449)

3. The paper discusses differences in F/G-actin ratio and cell cycle arrest but does not delve into potential signaling pathways (e.g., Src, PI3K/Akt, or p53 involvement). Authors might be performed the Western blot or RT-PCR analysis to provide depth to the mechanistic interpretation.

(Reply)

Thank you for your valuable comment. To investigate the signaling pathways, we conducted Western blot analysis and have added the corresponding methods, results, and discussion to the relevant sections of the manuscript. (line 191-201, 318-341, 425-440)

4. Statistical analysis performed again. The sample sizes (n=3 or n=4) are standard but rather small for robust conclusions. Increasing replicates or clarifying if experiments were performed in triplicate with separate biological repeats would be advisable.

(Reply)

Thank you for your valuable comment. As you rightly pointed out, having independent biological replicates would indeed strengthen the reproducibility of the findings. However, similar results were obtained in preliminary experiments conducted independently of the present study, which we believe supports the reproducibility of our findings. We hope for your kind understanding in this regard.

5. While the manuscript states that ARP and OPC are more effective than temozolomide (TMZ), a direct statistical comparison and dose equivalency discussion are missing. Therefore, authors added clarifying clinical relevance would enhance impact.

(Reply)

Thank you for your valuable comment. While our manuscript refers to the greater efficacy of ARP and OPC compared to temozolomide (TMZ), we acknowledge the importance of direct statistical and dose-equivalency comparisons. TMZ remained above 50�％ cell viability even at the highest tested concentration (300 µM), so IC₅₀ value of TMZ could not be precisely determined in our assay system. We have added this clarification to the revised manuscript (line 354-357)

6. The combined effect of DOX and ARP/OPC is shown, but synergy quantification (e.g., Combination Index or Bliss analysis) would provide stronger evidence.

(Reply)

Thank you for your valuable comment. We conducted a Bliss analysis to evaluate the combination effects, which quantitatively demonstrated synergistic interactions. We have added a discussion based on this analysis to the revised manuscript. (line 183-189, 306-310, 419-424)

7. The manuscript would benefit from additional language polishing for fluency and scientific tone.

(Reply)

We thank the reviewer for the suggestion regarding language and tone. In preparation for submission, the manuscript was professionally edited by a native English-speaker to ensure clarity, fluency, and an appropriate scientific tone. Nevertheless, we carefully re-reviewed the manuscript once again and made minor refinements where appropriate. We hope that the revised version now meets the language standards expected for publication.

8. Ensure all figures are high-resolution with clear legends. The figure captions should fully explain the experimental conditions.

(Reply)

Thank you for your valuable comment. We have revised the resolution and legends of the figures, and have also added more detailed experimental conditions to each figure caption.

9. Please define all abbreviations at first mention, including IM-54.

(Reply)

Thank you for your valuable comment. We have reviewed and revised the definitions of abbreviations throughout the manuscript. However, IM-54 is a proper name of a compound, so we have retained its original notation. We hope for your kind understanding.

10. Explain the rationale for choosing specific ARP and OPC concentrations, especially in combination studies.

(Reply)

Thank you for your valuable comment. We conducted combination experiments at 30 µM; however, no clear synergistic effects were observed under this condition. In contrast, more pronounced synergistic effects were seen at the 5 µM combination condition. Therefore, we have chosen to present the 5 µM data in this study as it more clearly demonstrates the combination effect. We have added a discussion of this point in the Discussion section of the revised manuscript.

11. Some references are repeated multiple times with different links. Ensure consistency and proper formatting.

(Reply)

Thank you for your valuable comment. We have carefully reviewed and revised the reference information.

---

## [Decision Letter · Decision Letter 1]

30 Jul 2025

Dear Dr. Shimizu,

Thank you for submitting your manuscript to PLOS ONE. After careful consideration, we feel that it has merit but does not fully meet PLOS ONE’s publication criteria as it currently stands. Therefore, we invite you to submit a revised version of the manuscript that addresses the points raised during the review process.

We look forward to receiving your revised manuscript.

Kind regards,

Mária A. Deli, M.D., Ph.D.

Academic Editor

PLOS ONE

Journal Requirements:

Reviewers' comments:

Reviewer's Responses to Questions

**Comments to the Author**

Reviewer #4: (No Response)

2. Is the manuscript technically sound, and do the data support the conclusions?

Reviewer #4: Yes

3. Has the statistical analysis been performed appropriately and rigorously?

Reviewer #4: I Don't Know

4. Have the authors made all data underlying the findings in their manuscript fully available?

Reviewer #4: Yes

5. Is the manuscript presented in an intelligible fashion and written in standard English?

Reviewer #4: Yes

Reviewer #4: Lines 30 are incorrect. “ In addition, both enhanced the efficacy of doxorubicin (DOX), which exhibits broad anticancer effects by inhibiting p-glycoprotein “. Yes, doxorubicin is a p-gp inhibitor but the primary MOA is damage to DNA.

Line 40 is deceptive. There are several attributes of GB that make it treatment resistant. See the MDACT paper for a more complete list. Likewise line 41 is deceptive. GB has spread throughout the entire brain at the time of diagnosis. See the MDACT paper Cancers. 2022 May 23;14(10):2563. or Kilmister et al Biomedicines. 2022 Nov 21;10(11):2988., “An effective treatment of cancer may require a multi-target strategy with multi-step inhibition of signaling pathways that regulate CSCs and the TME, in lieu of the long-standing pursuit of a 'silver-bullet' single-target approach.”

Line 44 is error. TMZ does not “ readily” pass thru the BBB. 20% brain tissue level compared to serum is not “readily”. It is “somewhat penetrating” or simple state the % or brain:serum ratio with reference.

Line 52 is an error. Aripiprazole is a D2 moderator. It is is both a partial agonist and a partial antagonist. This is an important distinction. See the data on dopamine agonism as growth enhancing in GB. As a side note here, the word “moderator” has been commonly misused in the medical literature in the last decade. It is commonly misused to indicate “reduced”. Correct use is “to bring toward the middle, or simply changed” as in AM or FM radio. Aripiprazole does bring D2 signaling towards the middle, blunting both high domaminergic signaling and the abnormally low signaling at D2 that is often associated with older antipsychotic medicines.

See the AVRO method of treating GB f

Line 54. Repurposing drugs for GB treatment, authors’ refs.6,7,8, should have more recent references. There have been five overviews in already by mid-year 2025, of the subject of repurposing already-marketed drugs for GB treatment [Starker et al, Int J Pharm. 2025 Jul 10:125935.; Kast et al, Int J Mol Sci. 2025 Jun 26;26(13):6158.; Gonzalez et al, Brain Sci. 2025 Jun 13;15(6):637,; De Silva et al, Trends Pharmacol Sci. 2025 May;46(5):392-406.,; Anwer et al, Clin Exp Med. 2025 Apr 13;25(1):117.]

Line 59, the examples given are arbitrary. The list is longer than that.

Line 61, “decrease” from what ? Better to say simply “are often different from the original drug as ingested”. Sometimes it is mainly the metabolite that is active, the originally ingested drug is inactive. Sometimes the original drug and its metabolites have completely different effects. Etc.

It is acceptable to abbreviate aripiprazole as the authors have done, but my preference is not to abbreviate aripiprazole. Remember readers will go thru many articles in a day’s study so minimizing abbreviations will help them.

For clarity Fig 4 A and Fig 4 B can, and should, be combined into a single bar graph as the authors have done in Fig 6 D. Also Fig 6 D must be made clearer. On my computer the two shades of grey appear too closely similar for designating aripip vs OPC. make on cross attached or otherwise clearly distinguishable. Same for Fig 7.

The authors must point out , emphasize, the dose discrepancy of aripiprazole and OPc between previous graphs that used 30 microM compared to Fig 7 studies that used 5 microM.

CYP2D6 and CYP3A4. Must be mentioned as the primary CYP forms responsible for OPC formation.

**Do you want your identity to be public for this peer review?** For information about this choice, including consent withdrawal, please see our Privacy Policy

Reviewer #4: No

---

## [Author Response · Author response to Decision Letter 2]

12 Sep 2025

Reviewer #4:

Lines 30 are incorrect. “In addition, both enhanced the efficacy of doxorubicin (DOX), which exhibits broad anticancer effects by inhibiting p-glycoprotein “. Yes, doxorubicin is a p-gp inhibitor but the primary MOA is damage to DNA.

reply)

Thank you for your valuable comment, and we apologize for the wording that could have been misleading. We have revised the sentence to read: “ARP and OPC enhanced DOX efficacy, likely via P-glycoprotein inhibition; known for ARP, suggested for structurally similar OPC.” (line 32, 33)

Line 40 is deceptive. There are several attributes of GB that make it treatment resistant. See the MDACT paper for a more complete list. Likewise line 41 is deceptive. GB has spread throughout the entire brain at the time of diagnosis. See the MDACT paper Cancers. 2022 May 23;14(10):2563. or Kilmister et al Biomedicines. 2022 Nov 21;10(11):2988., “An effective treatment of cancer may require a multi-target strategy with multi-step inhibition of signaling pathways that regulate CSCs and the TME, in lieu of the long-standing pursuit of a 'silver-bullet' single-target approach.”

reply)

Thank you for this important point. We have revised the text to emphasize that resistance is multifactorial (CSCs, TME, intratumoral heterogeneity, adaptive/compensatory signaling) and that diffuse, brain-wide infiltration at diagnosis limits curative resection. We also aligned our wording with the literature advocating multi-target, multi-step strategies rather than a single-target “silver-bullet” approach, and we have added the suggested references. (line 43-49)

Line 44 is error. TMZ does not “readily” pass thru the BBB. 20% brain tissue level compared to serum is not “readily”. It is “somewhat penetrating” or simple state the % or brain:serum ratio with reference.

reply)

Thank you for the helpful comment. We have revised the text to read: “Although TMZ partially crosses the blood–brain barrier, its concentration in cerebrospinal fluid is approximately 20% of that in plasma.”

(line 51, 52)

Line 52 is an error. Aripiprazole is a D2 moderator. It is is both a partial agonist and a partial antagonist. This is an important distinction. See the data on dopamine agonism as growth enhancing in GB. As a side note here, the word “moderator” has been commonly misused in the medical literature in the last decade. It is commonly misused to indicate “reduced”. Correct use is “to bring toward the middle, or simply changed” as in AM or FM radio. Aripiprazole does bring D2 signaling towards the middle, blunting both high domaminergic signaling and the abnormally low signaling at D2 that is often associated with older antipsychotic medicines.

reply)

Thank you for the valuable comment. As suggested, we have revised the description of aripiprazole from “a high-affinity partial dopamine D2 receptor agonist” to “a D2 receptor modulator.” (line 66)

See the AVRO method of treating GB

reply)

Thank you for the valuable comment. We have added a mention of the AVRO regimen in the Introduction and included the relevant references. (line 64, 65)

Line 54. Repurposing drugs for GB treatment, authors’ refs.6,7,8, should have more recent references. There have been five overviews in already by mid-year 2025, of the subject of repurposing already-marketed drugs for GB treatment [Starker et al, Int J Pharm. 2025 Jul 10:125935.; Kast et al, Int J Mol Sci. 2025 Jun 26;26(13):6158.; Gonzalez et al, Brain Sci. 2025 Jun 13;15(6):637,; De Silva et al, Trends Pharmacol Sci. 2025 May;46(5):392-406.,; Anwer et al, Clin Exp Med. 2025 Apr 13;25(1):117.]

(Reply)

Thank you for the valuable comment. Among the references you suggested, we have cited those most pertinent to our study.

Line 59, the examples given are arbitrary. The list is longer than that.

(Reply)

Thank you for pointing out that our examples could be construed as arbitrary. To avoid implying exhaustiveness, we revised the sentence to present the tumor types as representative examples. (line 67-69)

Line 61, “decrease” from what? Better to say simply “are often different from the original drug as ingested”. Sometimes it is mainly the metabolite that is active, the originally ingested drug is inactive. Sometimes the original drug and its metabolites have completely different effects. Etc.

(Reply)

Thank you for the valuable comment. As suggested, we have revised the text for clarity to read: “Most drugs are metabolized in vivo and the effects of these metabolites are often different from the original drug as ingested.” (line 73-75)

It is acceptable to abbreviate aripiprazole as the authors have done, but my preference is not to abbreviate aripiprazole. Remember readers will go thru many articles in a day’s study so minimizing abbreviations will help them.

(Reply)

Thank you for this helpful suggestion regarding abbreviations. We agree that excessive abbreviations can hinder readability. Because “aripiprazole” and “OPC-14857” appears frequently throughout the manuscript, replacing all instances with the full name would substantially lengthen sentences and figure legends. To balance clarity and readability, we have retained the abbreviation ARP, while (i) spelling out “aripiprazole (ARP)”, “OPC-14857 (OPC)”at its first occurrence in this article, (ii) minimizing the introduction of any additional abbreviations, and (iii) ensuring consistent usage across the text and figure captions. We hope this compromise addresses the concern; if the Editor prefers, we are happy to replace ARP with the full name throughout.

For clarity Fig 4 A and Fig 4 B can, and should, be combined into a single bar graph as the authors have done in Fig 6 D. Also Fig 6 D must be made clearer. On my computer the two shades of grey appear too closely similar for designating aripip vs OPC. make on cross attached or otherwise clearly distinguishable. Same for Fig 7.

(Reply)

Thank you for the valuable comment. With respect to the bar graphs in Figure 4, they were originally presented as a single combined panel; however, at an earlier stage a different reviewer requested that we split them. In light of these conflicting recommendations, we re-discussed the presentation within our group and have now reinstated the combined format. For Figures 6D and 7, we adopted distinct fill patterns to improve visual clarity and distinguishability between conditions.

The authors must point out, emphasize, the dose discrepancy of aripiprazole and OPc between previous graphs that used 30 microM compared to Fig 7 studies that used 5 microM.

(Reply)

Thank you for your valuable comment. We conducted combination experiments at 30 µM; however, no clear synergistic effects were observed under this condition. In contrast, more pronounced synergistic effects were seen at the 5 µM combination condition. Therefore, we have chosen to present the 5 µM data in this study as it more clearly demonstrates the combination effect. We have added a discussion of this point in the Discussion section of the revised manuscript. (line 422-425)

CYP2D6 and CYP3A4. Must be mentioned as the primary CYP forms responsible for OPC formation.

(Reply)

Thank you for the valuable comment. In the Introduction, we have added a statement noting that OPC is a metabolite of aripiprazole formed by CYP2D6 and CYP3A4. (line 78, 79)

---

## [Decision Letter · Decision Letter 2]

25 Sep 2025

Dear Dr. Shimizu,

Thank you for submitting your manuscript to PLOS ONE. After careful consideration, we feel that it has merit but does not fully meet PLOS ONE’s publication criteria as it currently stands. Therefore, we invite you to submit a revised version of the manuscript that addresses the points raised during the review process.

We look forward to receiving your revised manuscript.

Kind regards,

Mária A. Deli, M.D., Ph.D.

Academic Editor

PLOS ONE

Journal Requirements:

Reviewers' comments:

Reviewer's Responses to Questions

**Comments to the Author**

Reviewer #4: (No Response)

2. Is the manuscript technically sound, and do the data support the conclusions?

Reviewer #4: Yes

3. Has the statistical analysis been performed appropriately and rigorously?

Reviewer #4: I Don't Know

4. Have the authors made all data underlying the findings in their manuscript fully available?

Reviewer #4: Yes

5. Is the manuscript presented in an intelligible fashion and written in standard English?

Reviewer #4: Yes

Reviewer #4: Aripiprazole

This is a well written addition of average interest and importance to glioblastoma researchers. I recommend publishing it only after mention of, and discussion of, the several recent reviews on the general nature of D2 blocking or D2 binding drugs’ inhibition of glioblastoma growth.

Line 59. It is important to mention here the several recent discussions on the general nature of all drugs that bind to the D2 receptor having inhibitory effects on GB growth. See several publications in the last year on olanzapine listing 27 FDA-EMA-approved D2 binding drugs for treating psychosis. Of these 27 currently marketed antipsychotic drugs, 24 have evidence of GB growth inhibition similarly to apripiprazole. A total database on this has over 84 different studies. These recent reviews of D2 inhibition in glioblastoma must be discussed in the Discussion section as well.

Data on perphenazine inhibition of glioblastoma must be mentioned. Perphenazine is a D2 binding antipsychotic drug that achieves higher brain tissue levels than does aripiprazole.

If the authors wanted to discuss putative mechanisms of action for D2 antagonists’ action inhibiting glioblastoma, that would be a welcome addition to their paper.

**Do you want your identity to be public for this peer review?** For information about this choice, including consent withdrawal, please see our Privacy Policy

Reviewer #4: No

---

## [Author Response · Author response to Decision Letter 3]

10 Oct 2025

Reviewer #4: Aripiprazole

This is a well written addition of average interest and importance to glioblastoma researchers. I recommend publishing it only after mention of, and discussion of, the several recent reviews on the general nature of D2 blocking or D2 binding drugs’ inhibition of glioblastoma growth.

Line 59. It is important to mention here the several recent discussions on the general nature of all drugs that bind to the D2 receptor having inhibitory effects on GB growth. See several publications in the last year on olanzapine listing 27 FDA-EMA-approved D2 binding drugs for treating psychosis. Of these 27 currently marketed antipsychotic drugs, 24 have evidence of GB growth inhibition similarly to apripiprazole. A total database on this has over 84 different studies. These recent reviews of D2 inhibition in glioblastoma must be discussed in the Discussion section as well.

Data on perphenazine inhibition of glioblastoma must be mentioned. Perphenazine is a D2 binding antipsychotic drug that achieves higher brain tissue levels than does aripiprazole.

If the authors wanted to discuss putative mechanisms of action for D2 antagonists’ action inhibiting glioblastoma, that would be a welcome addition to their paper.

reply)

Thank you for the valuable comment. We agree that discussion of the D2 receptor is an important point. We have added a dedicated paragraph on D2-receptor–related mechanisms in the Discussion and have cited relevant studies on perphenazine. (line 454-469)

---

## [Editor Report · Decision Letter 3]

30 Nov 2025

Effect of anticancer activity of aripiprazole main metabolite OPC-14857 on malignant glioblastoma.

PONE-D-24-44585R3

Dear Dr. Shimizu,

We’re pleased to inform you that your manuscript has been judged scientifically suitable for publication and will be formally accepted for publication once it meets all outstanding technical requirements.

Kind regards,

Mária A. Deli, M.D., Ph.D.

Academic Editor

PLOS ONE

Additional Editor Comments (optional):

The authors responded to the last minor comment of Reviewer 4, and added the requested paragraph to the Discussion and 4 references. All the comments were addressed.
---

## [Editor Report · Acceptance letter]

PONE-D-24-44585R3

PLOS One

Dear Dr. Shimizu,

I'm pleased to inform you that your manuscript has been deemed suitable for publication in PLOS One. Congratulations! Your manuscript is now being handed over to our production team.

Kind regards,

on behalf of

Prof. Mária A. Deli

Academic Editor

PLOS One